health and disease and epidemiology, microbiology, behaviour

COVID-19, foot-and-mouth disease, pandemic threats, exit strategies, biosecurity, emergency management

**Author for correspondence:**
Keith Sumption
e-mail: keith.sumption@fao.org

# Parallels, differences and lessons: a comparison of the management of foot-and-mouth disease and COVID-19 using UK 2001/2020 as points of reference

Keith Sumption[1], Theodore J. D. Knight-Jones[2], Melissa McLaws[1] and David J. Paton[1]

[1]European Commission for the control of Foot-and-Mouth Disease, via Terme di Caracalla, Rome, Italy
[2]International Livestock Research Institute (ILRI), Arusha, Tanzania

 KS, 0000-0002-3638-9869; DJP, 0000-0002-9097-2262

Foot-and-mouth disease (FMD) is an extremely infectious viral infection of cloven-hoofed animals which is highly challenging to control and can give rise to national animal health crises, especially if there is a lack of pre-existing immunity due to the emergence of new strains or following incursions into disease-free regions. The 2001 FMD epidemic in the UK was on a scale that initially overwhelmed the national veterinary services and was eventually controlled by livestock lockdown and slaughter on an unprecedented scale. In 2020, the rapid emergence of COVID-19 has led to a human pandemic unparalleled in living memory. The enormous logistics of multi-agency control efforts for COVID-19 are reminiscent of the 2001 FMD epidemic in the UK, as are the use of movement restrictions, not normally a feature of human disease control. The UK experience is internationally relevant as few countries have experienced national epidemic crises for both diseases. In this review, we reflect on the experiences and lessons learnt from UK and international responses to FMD and COVID-19 with respect to their management, including the challenge of preclinical viral transmission, threat awareness, early detection, different interpretations of scientific information, lockdown, biosecurity behaviour change, shortage of testing capacity and the choices for eradication versus living with infection. A major lesson is that the similarity of issues and critical resources needed to manage large-scale outbreaks demonstrates that there is benefit to a 'One Health' approach to preparedness, with potential for greater cooperation in planning and the consideration of shared critical resources.

## 1. Introduction

Foot-and-mouth disease (FMD) is a highly infectious viral disease with a massive economic impact on livestock sectors and livelihoods where it is endemic (much of Africa and Asia) and through outbreaks in FMD-free countries (Europe, most of the Americas, parts of southern Africa and Oceania) due to production losses, costs of control and trade restrictions [1]. Pandemics of FMD are prevented by restricting movements of animals, but at least twice in the last 20 years, peridemics (regionally restricted multi-country epidemics) have spread rapidly to other regions. In the case of the FMD serotype O Pan-Asia strain, the extensive amplification and spread in 1999–2000 presaged its entry to the UK and northern Europe in 2001 [2]. In 2020, the rapid emergence of coronavirus disease 2019 (COVID-19) has led to a global human pandemic that is having profound impacts on our way of life. Comparison of the emergence and management of COVID-19 and FMD in the UK in 2001 (FMD2001) is particularly apt, given the similarities in disease transmission

and crisis management approaches and challenges. There are also globally relevant similarities in the control measures applied, many of which are routinely used in animal health but rarely for human diseases in recent times (restrictions on movement and mixing, and enhanced biosecurity). Given this, many of the lessons learnt from controlling FMD are relevant to COVID-19.

## 2. Foot-and-mouth disease virus and SARS-CoV-2: comparative viral and transmission characteristics

This paper focuses on parallels in disease management, but it is useful to compare underlying transmission characteristics of the two viruses. FMD is caused by an RNA virus in the Picornavirus family with a high mutation rate, seven sero-types and periodic emergence of new strains. It is highly infectious, with aerogenic infection as the main route of transmission between ruminants in direct contact. Indirect transmission also occurs via fomites and abrasions and by ingestion, pigs being more readily infected than ruminants by the latter route [3]. Longer distance airborne spread between farms occurs relatively rarely, mostly from pigs to cattle. The relative importance of direct and indirect transmission modes is difficult to quantify [4], but significant farm-to-farm spread can continue after imposition of complete animal movement standstills, particularly in areas of high farm density [5]. Viral characteristics underlying the explosive spread of the FMDV O PanAsia topotype in 1999–2001, and its apparent ability to out-compete type O strains circulating in regions it entered have been investigated but not identified [6]. Like FMDV, COVID-19 is also highly contagious and thought to be transmitted by a combination of respiratory droplets/aerosols and fomites; however, significant knowledge gaps remain (see electronic supplementary material, table S1) [7].

## 3. FMD2001 and COVID-19: scale and impacts

FMD was first suspected in the UK in 2001 at an abattoir in Essex, on 19 February 2001, and by the time the disease had been eradicated at the end of September 2001, over six million animals had been culled as well as an unknown number of youngstock [8], comprising roughly 12% of all UK farm livestock (electronic supplementary material). Over 2,000 premises (1% of susceptible holdings [9]) were infected across 44 administrative divisions, and in mid-April 2001, at the height of the crisis, well over 10 000 vets, soldiers, and field and support staff were engaged in fighting the disease, and up to 100 000 animals were slaughtered and disposed of each day [10]. Tourism suffered the largest financial impact from the outbreak, with visitors to Britain and the countryside deterred by the initial blanket closure of footpaths by local authorities and media images of mass pyres.

As of 12th June 2020, official figures for laboratory confirmed UK cases of SARS-CoV-2 stand at 292 950 (0.4% morbidity), with 41 481 (0.06% mortality) COVID-19-associated UK deaths [11]. Serological studies, as conducted after FMD2001 and which peaked at 220 000 tests/week [12], will be key to understanding the level of SARS-CoV-2 exposure of the population, which, like FMD in the UK in

2001, had no pre-existing level of immunity. Arguments have been made that more years of life will be lost due to the subsequent recession than will be gained through beating the virus [13], which mirrors an argument that the impact of measures used in 2001 far exceeded the actual clinical impact of FMD [14]. It could be argued that both livestock community and the public in 2001 largely had no recent memory of FMD and therefore were bewildered by the sudden descent to quarantines and 'medieval' forms of 'plague' control, and a similar parallel is found in COVID-19.

## 4. Preparedness

### (a) Before the imminent threat is perceived

#### (i) Prevention of international spread

FMD-free countries such as the UK have a raft of measures, continuously in place, to prevent incursions of FMD virus and other key animal pathogens (figure 1). In the case of FMD, disease-free status is very important for international trade in susceptible livestock and their products, such as meat and milk. As a consequence of the measures taken, FMD outbreaks in disease-free regions are infrequent—usually less than five incursions per year globally and most begin as a single point of introduction [16]. This contrasts markedly with COVID-19, for which, as a new disease, there were no pre-existing measures and where multifocal introductions have occurred in UK (more than 1000 separate introductions) and many other newly affected countries [17]. The differences between routine measures taken to prevent incursions of FMD and COVID-19 mirror the situation more generally between livestock and people in terms of preventing the spread of infectious diseases. Whereas livestock movements between countries are often tightly regulated according to health status [15], very few such controls are usually in place for the movement of people, and with the exception of travel to regions with risks of yellow fever or Ebola, few restrictions and health controls are enforced, regardless of the return country. This rapidly changed as the COVID-19 outbreak evolved with restrictions placed on international travel and/or requirements for quarantine. The evolving development of agreements on travel between countries is beginning to mirror the international set of protocols for the safe movement of livestock between countries of different animal health status.

### (ii) Understanding and mitigating the key transmission events behind overwhelming outbreaks

To identify and mitigate the key risk factors that can transform a small outbreak into a large one is critical when preventing overwhelming FMD outbreaks [18]. Risk reduction measures taken included banning livestock markets (e.g. The Netherlands [19]) and feeding swill to pigs (EU-wide ban). After 2001, the UK continued with live animal markets but prohibited animals from leaving premises where the new stock had arrived for a certain time to reduce undetected spread [20]. Key early transmission events driving large national COVID-19 epidemics are not yet clear, although a market in Wuhan, China, a church service in South Korea [21] and a football match in February in Italy have been proposed in the respective countries [22]. Although the mechanisms behind the emergence of COVID-19 are not yet

**Figure 1.** FMD controls in relation to country FMD status recognized by the World Organisation for Animal Health [15]. [1]FMD-free zones are also recognized for the purposes of trade, based on geographic separation of livestock sub-populations. [2]Slaughter and disposal of affected herds followed by cleansing and disinfection and controlled restocking. [3]Can also cease vaccination and move directly to free without vaccination. [4]Endemic countries are those without FMD freedom officially recognized by the World Organisation for Animal Health (OIE). (Online version in colour.)

understood, the importance of live animal markets in virus emergence has long been recognized [7]. Keeping animals of different species together, especially wildlife species, within food markets can facilitate the emergence of zoonoses [23,24].

### (iii) Preparedness planning for large-scale epidemics

In the aftermath of FMD2001, the National Audit Office (NAO) inquiry concluded that the UK national contingency plan, based on the most likely scenario of 10 infected premises, had been inadequate, and recommended that plans should incorporate a range of different assumptions about the nature, size and spread of an outbreak and have regard to the wide economic, financial and environmental impacts of different methods of disease control [8] (table 1). Similarly, the extent of the COVID-19 outbreak has far exceeded the expected scale of an emerging infectious disease, foreseen by many to be similar to that of SARS in Toronto, with 251 cases over several months [29]. A cognitive bias towards the scenarios of seasonal influenza pandemics has been postulated in the initial UK response to COVID-19, assuming that it will behave like the outbreak you have prepared for [30]. However, established control and treatment options for influenza, such as vaccination and anti-viral medications, are/were not available for COVID-19, necessitating contact tracing to identify cases and prevent spread, a cornerstone of FMD control.

COVID-19 and FMD2001 have both been national crises in which the assistance of the military and private sector were brought in when it appeared national public and animal health capacities were at risk of being overwhelmed. It is essential to estimate and assess the impacts of changes in the availability of critical human resources and key capabilities *a priori*. More complete modelling approaches are needed that include the health impacts, critical health system capacities and resources. Such models have been built for FMD [10], but similar 'whole system' models are

urgently needed for emerging pathogens and must have an acceptable level of validation for use in 'real-time' emergency settings.

### (b) Response to the imminent threats

International concerns and warnings had been raised [31] in the months before the UK FMD2001 incursion, following the rolling wave of O Panasia epidemics from 1998 to 2000; in late November 2000, countries were warned that they should 'recognize the deterioration of the FMD situation worldwide and that they should reappraise their strategies and operations' [31]. A possible cognitive bias may have blunted the response to warnings, since expert opinion at that time considered FMD risks to the EU to be mainly related to land border entry from Turkey [31].

With COVID-19, in the first three weeks of 2020, 17 flights from Wuhan and over 600 from the rest of China landed in the UK [30], and the disease had already spread to other countries in the region, reaching Europe, the Middle East, Russia, Australia and North America by the end of January, at which point it was declared a public health emergency of international concern by WHO [32]. In fact, COVID-19 is thought to have been introduced to the UK over 1300 times, the vast majority resulting from infected people travelling from Europe, mostly between late February and late March 2020 [17].

FMD2001 and COVID-19 both highlight the significance of mass human and animal transit in highly developed economies with globalized human and animal product traffic, and the limited information and time available on which to risk assess new events of potential global significance. Of common concern is the performance of surveillance systems and transparency in reporting, and to maintain these, the need to avoid punitive counter measures that act as disincentives to international reporting. Information on the movements of livestock and where they are kept are routinely collected in

**Table 1.** Lessons from FMD2001 selected[a] for possible relevance to COVID-19.

| early warning | need to strengthen early warning systems [25] |
| --- | --- |
| precautionary principle | use of the precautionary principle to ensure outbreaks do not become epidemics including need to consider national lockdown when first case detected and until extent of spread clear [8,25] |
| leadership | although ministers and their veterinary experts and officials recognized from the outset that they were facing a serious situation, no one in command understood in sufficient detail what was happening on the ground during these early days [26] |
| contingency plans | contingency plans should (1) prepare for worst case scenarios as well as most likely ones; (2) cover crisis procurement of personnel, goods and services; (3) be regularly reviewed and tested [8,25] |
| testing | need for rapid testing and reporting including point-of-care tests [25] |
| science | scientific experts must be accountable, not only to government ministers but also to other experts [27] |
| modelling | policy mistakes can arise from inappropriate use of epidemiological models [27] and need for resource modelling to assess feasibility of options [10] |
| centralization | value of decentralization and local expertise and practice [28] |

[a]The selection is not a comprehensive representation of the many recommendations, enquiries, publications and revised rules and guidelines in the aftermath of FMD 2001.

the EU, especially for movements between countries [33]. In 2001, the UK confirmation of FMD triggered the EU system for surveillance, tracing and immediate control of animals received in the weeks before confirmation. This tracing greatly reduced the European FMD epidemic, although Ireland, France and the Netherlands were affected due to movements of animals that took place prior to the detection of FMD in the UK [8]. For COVID-19, triggers for activating defined pandemic control measures may not have been adequately established in advance, and possibly led to critical delays in disease control. Recognition of the unusual severity of the disease and the impact this would have in terms of critical care capacity appear delayed, despite the fact that the UK was not the first European country affected; COVID-19 was considered only a 'moderate' threat to the UK until 12 March (figure 2).

## 5. Management of FMD2001 and COVID-19

### (a) Phase 1: attempted containment

#### (i) Rapid detection

Multiple failures in recognition of disease and reporting occurred before FMD was recognized in the UK in 2001 and this delay had a decisive effect upon the capacity to contain outbreaks (electronic supplementary material) with at least 57 farms having been infected across the UK before the disease was confirmed [34]. As few as 16 infected sheep, taken first to Hexham market, and nine of which were sold again days later at the Longtown market, resulted in the national dispersion of FMD to around 79 holdings across the UK [35]. With COVID-19, the potential for transmission during brief contact periods was initially considered minimal if the person was not clinically ill, and delays in the closure of schools and public events have been a major controversy [22]. Common to both FMD and COVID-19 are asymptomatic, relatively mild or unreported cases, but for COVID-19 these began to be recognized only in late February on the basis of tracing and testing in-contact persons [36]. In 2001, the scale of tracing required rapidly overwhelmed veterinary capacity in many

regions. Laboratory test capacity was under enormous pressure immediately and, under targets for culling introduced in March, slaughter of both affected and in-contact farms proceeded often without laboratory testing. An immediate clamour for 'pen-side' test devices was made by livestock owners, based on reports of portable tests developed in the US [37] and these were examined for utility but never applied [38]. However, samples were collected from 85% of outbreak premises, allowing retrospective analysis that could not confirm virus or antibodies in 23% of the slaughtered premises [39]. Since 2001, the development of point-of-care (POC) testing for FMD has been a priority [25] and RT-LAMP tests for FMD were developed and validated [40].

The lack of test capacity has also hampered confirmation of SARS-CoV-2 involvement in mortalities. The problems in the application of high-throughput tests for FMD and POC devices mirror the issues with COVID-19. Some of the same scientists have assisted in the validation of analogous methods for COVID-19 now undergoing trials in UK hospitals [41]. Intensive, pre-clinical surveillance sampling using RT-PCR for FMD virus enabled tighter control of the 2007 epidemic in the UK [42] and high-throughput test-and-trace systems for COVID-19 that were established quickly in 2020 have been credited with early success in containment in several countries [43]. In the UK in 2020, the initial policy was to centralize mass testing rather than use existing capacity in a range of sectors, including veterinary laboratories [44]. National capacities to ramp up test throughput are thus shown to be critical in both human and animal emergency management, often using the same platforms, and national preparedness planning should therefore consider if a common resource pool and ramping up capacity could efficiently meet both types of need.

#### (ii) Community transmission and the timing of movement restrictions

Tight control of animal movements has been a hallmark of British FMD control procedures since the 1890s, with 'lockdowns' rigorously and immediately applied at farm level,

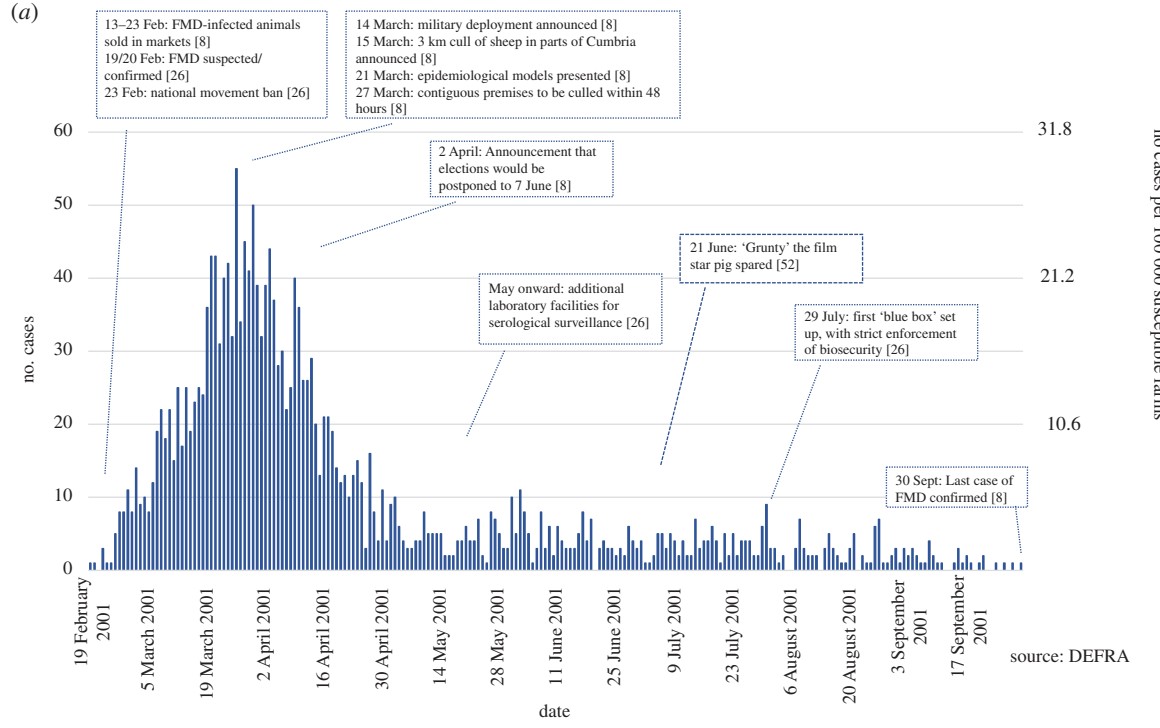

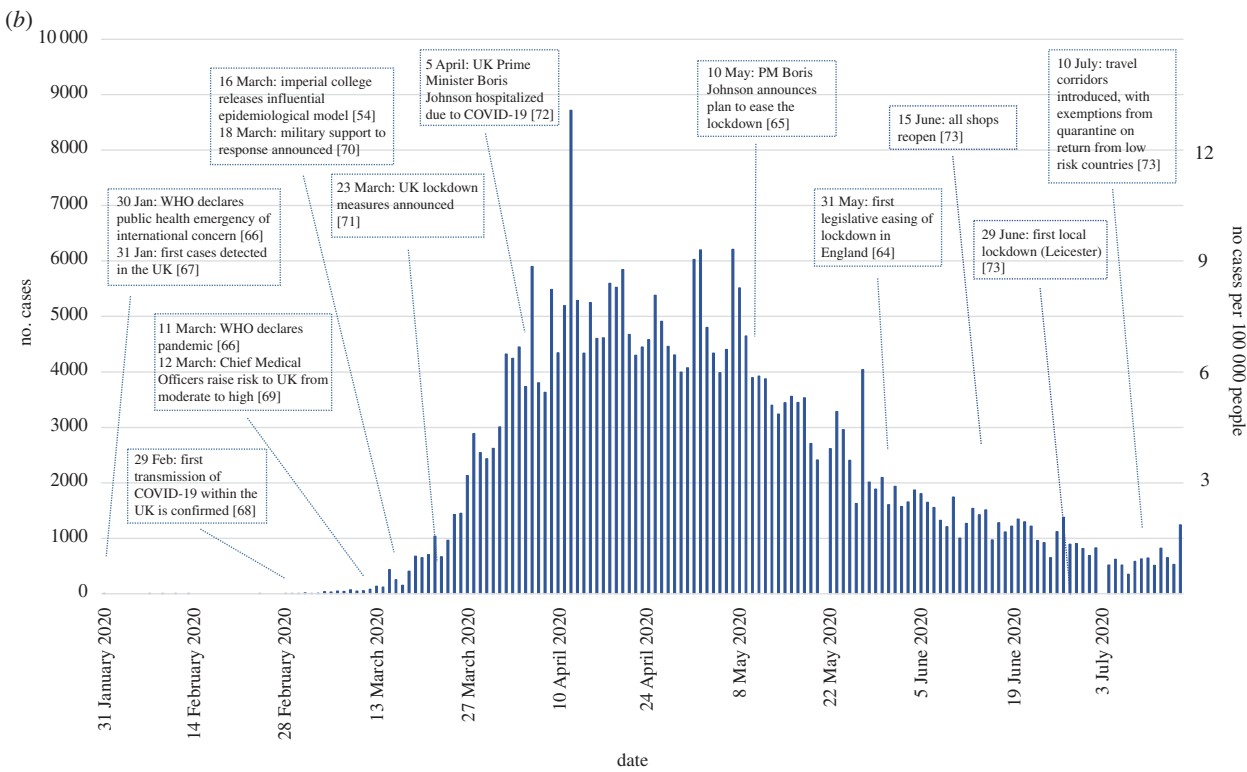

Source: [74] EU Open Data Portal. Data adjustments took place on 21 May (−525 cases) and 3 July (−29 726 cases)

**Figure 2.** Annotated epidemic curves of the 2001 outbreak of FMD (*a*) and COVID-19 in 2020 in the UK (*b*), additional references in electronic supplementary material. (Online version in colour.)

and vigorous tracing operations. International movements of susceptible animals and their products are suspended, and in animal health, unlike human health, the EU member states cooperate in a system of shared consignment health status information and apply centrally agreed controls on movements of animals and animal products across borders [33]. The UK control strategy for COVID-19, was, initially like FMD2001, to 'contain' infection but switched to 'delay [spread]' on 13 March, with the initial indication of acceptance of transmission as necessary to establish an eventual

herd immunity; this was radically changed to lockdown measures within a week [22]. This delay is now thought to have at least doubled the number of UK COVID-19 deaths to date and in-part resulted from a failure of surveillance systems to detect the scale of virus incursions and transmission at that time [45].

In the UK in 2001, after detecting the FMD outbreak there was a critical four-day delay from the time that FMD was first suspected on the morning of 19 February until a national lockdown (movement standstill) was implemented late in

the afternoon of 23 February. It was estimated that if the national movement ban had been imposed on 20 February, the day the disease was confirmed; the final epidemic would have been reduced by one third to one half [26]. However, this delay is much shorter than the 21 days between detection of the first UK COVID-19 case and lockdown, partly explaining the exponential epidemic growth seen for COVID-19 but not FMD2001. The UK's FMD control strategy currently stipulates that a national movement ban will be implemented at the beginning of any FMD outbreak [46]. A major challenge for COVID-19 control will be to define the trigger points and extent of 'lockdowns' in relation to the transmission situation in the country, geographic area or at-risk groups of the community.

For FMD, it must be noted that movement restriction (isolation of infected farms) has almost never been sufficient by itself to prevent transmission. 'Over the fence' transmission to contiguous premises in 2001 was a rationale for the increasingly aggressive culling strategies introduced in March 2001 (figure 2). Further details of FMD control measures in UK in 2001 are given in the electronic supplementary material.

## (b) Phase 2: revisions in strategy and the role of scientific guidance during early epidemic growth

The initial steep rise in reported FMD cases in 2001 was the result of the nationwide seeding of infection with cases detected rapidly due to heightened awareness and a huge effort in surveillance (assisted by a 600% increase in veterinary field staff). Daily outbreak update reports were national media events, and brought intense public scrutiny, as also was observed for COVID-19. Every rise in cases was seen as a failure to take firm and decisive actions, even though the effects of any action would not be seen for at least one incubation period [16]. Confidence in the ability of the traditional measures of movement restrictions and stamping out to achieve rapid containment came under severe pressure with the intensity of demand for firmer actions. Controversial at the time and afterwards was the use of scientific evidence and particularly that arising from disease spread modelling groups. Their projections informed decisions on unprecedented culling strategies and targets for the speed of culling operations (slaughter and disposal), and influenced public opinion on whether the epidemic was under control. Faced with the enormity of the culls, the military were engaged to manage the logistics of culling and disposal: mass burial sites were constructed in each region for the millions of carcasses that resulted from disease control (4.2 million) and culls for welfare reasons (2.3 million) as animals accumulated on farms without the possibility to be moved and grazing became exhausted [8]; in total about 7% of UK cattle and 15% of UK sheep were culled [1].

The accuracy of models used to inform policy in the UK in both epidemics has been questioned. In the case of FMD in 2001, the model results indicated that controls were insufficient and called for additional culling. However, new cases started to decline before these measures had time to take effect, suggesting that existing controls were already succeeding [27]. In 2020, the number of early cases of SARS-CoV-2 was underestimated due to a lack of testing but crude doubling times for observed deaths were three days and not the five days reported from Wuhan that were used to parameterize models. This may have contributed to underestimating

progression of the UK epidemic and delayed introduction of a national lockdown [47]. A lesson from FMD2001 was the need to have appropriate, validated models developed and tested before the crisis, not during it. For rapidly emerging pathogens like COVID-19, this is a challenge.

In both 2001 and 2020, 'dire' forecasts from the same modelling group appear to have had the greatest influence on decisions [8,48]. There is an urgent need to better understand how 'pessimistic' projections seem to drive decisions during emergencies yet fail to do so at other times when scenarios are considered for emergency preparedness. Modelling should be central to decisions on the management of national recovery from COVID-19, but given the trade-offs between health and economic outcomes, it requires policy makers, rather than scientists, to decide which outcomes are the priority. A lesson from 2001 that does not appear to be fully learnt is the need for greater openness on the range of model types, assumptions, parameters and outputs being used for guidance of decisions of enormous significance, as well as the relative weight of modelling and other guidance on national decisions on imposition of lockdowns. Over centralized decision-making and disregard for local knowledge and initiative was an issue for FMD control in 2001, for example, in culling decisions [28] and has been cited as causing delays and lack of testing, flexibility, innovation and capability when dealing with COVID-19 [49,50].

## (c) Phase 3: the height of the battle: the search for assurances at times of national crisis

In 2001, the battle for control of media and the narrative was intense, not least since the general election had to be postponed. Messages during FMD2001 bear striking parallels with the intense efforts to 'flatten the curve' and protect the UK health services in 2020. For the 2001 FMD epidemic, the period from 1st to 19th April 2001 was seen as the battle of the data and some argued that this was deliberate in order to 'get past the peak' ahead of the decision on the date of the postponed general election [51]. In COVID-19, there has been a similar concern over data being compared, as the denominators for test results vary by date and by country, and also whether non-confirmed cases should be counted to understand the effect on the total population mortality. Lack of testing and reporting has further clouded the issue. The reporting of 'glimmers of light' that the epidemic peak may have been reached appears common to both FMD and COVID-19 (electronic supplementary material), as were legal challenges to control measures. The farmer fight-back against the 'contiguous culls' grew from March 2001, but it took the High Court case sparing 'Grunty the film star pig' on 21 June 2001 [52] to significantly affect the application of automatic culls. However, by then, the scale of loss had been enormous.

## (d) Phase 4: exiting from foot-and-mouth disease and COVID-19 epidemics: comparisons

### (i) The need for consensus on the exit or long-term strategy matters even before the epidemic peaks

In both COVID-19 and FMD2001, exit or long-term strategies have been of critical importance for garnering public support for and compliance with control measures. For COVID-19, balancing the societal, economic and health prerogatives

through adaptive use of measures affecting transmission appears the common position of most countries that have passed the initial epidemic peak, but remain wary of further epidemic waves. An eradication scenario for SARS-CoV-2, as used in FMD2001, appears to be the strategy chosen by a few countries with low case numbers, such as New Zealand, which to a certain extent may be a result of their existing stringent import and border controls (motivated in part to exclude FMD) and their relative isolation, but like FMD-free countries, ongoing restrictions are then needed to prevent virus incursions and protect the highly susceptible population.

COVID-19 and FMD2001 both require/required mass diagnostic screening that was not present in the first months of each epidemic. In the FMD case, it was clear from early in March 2001 that to regain internationally recognized freedom for a trade would require mass serological screening. To achieve this, new tests had to be developed 'in-house' and validated under pressure to enable screening of 3.5 million sera in the summer and autumn 2001. In FMD2001, additional serology testing centres (including medical ones) were engaged, and results were critical to the UK regaining official freedom in January 2002. Since 2001, maintaining test capability has been an ongoing part of FMD contingency planning.

With COVID-19, a serological test for mass screening has also been identified by the WHO as a priority to disclose occult infection (as for FMD) as well as to measure population immunity (if immunity is protective) [53]. Such tests make possible consideration of exit strategies that include the build-up of seropositive individuals after recovery from infection and a reduction in movement restrictions, based on either levels of infection or immunity (personal, age-group or even area-based). However, herd immunity through natural infection would equate to a high disease burden, requiring infection and immunity in most of the population, and would take a long-time to develop, especially where transmission is well controlled. This suggests vaccination is required to achieve adequate population immunity [54]. Notwithstanding the value of serology, a high capacity for rapid virological testing to detect and trace SARS-CoV-2 cases remains essential for future COVID-19 control, whereas although its lack was felt in FMD2001, the virus was eradicated without it, as on previous occasions.

## (e) Phase V: post-epidemic reconstruction and long-term impacts

FMD2001 had profound impacts, socially, politically and in the long term on the health of humans and the livestock sectors. High levels of post-traumatic stress disorder (PTSD) were seen in farmers and PITS (perpetration induced traumatic stress) was coined to describe the impact on workers who had undertaken culling and disposal procedures [55]. As well as the many deaths, it is very likely that mental health issues will be felt for years by medical teams after COVID-19, who, like veterinarians, are largely not trained for a 'battlefield' intensity but have been working in situations where exceptional levels of mortality must be experienced, with a major risk to life and health in their front line operations.

The animal health impact of FMD2001 was felt for at least a decade afterwards, since restocking after culling severely worsened the national bovine tuberculosis control programme [56]. The need for economic recovery plans was evident in almost all reviews, but limitations on state aid were seen as preventing the UK government from assisting business recovery. There was a particular grievance that only 39 million British pounds (GBP) were provided for business recovery, whereas 1.34 billion GBP had been paid in compensation to farmers, and 5 years after FMD2001, the impacts remained on the economies of the worst-affected counties, Cumbria and Devon [57].

The non-COVID health impacts will be far more profound, from the consequences of medical procedures foregone through to the health impacts, positive and negative of isolations, and of the economic impacts on society. It is extremely unlikely that any civil disaster planning had the foresight to prepare a suitable reconstruction strategy for such a scenario in which the society is affected, and economies impacted, on a global scale.

## (i) The long-term management of COVID-19 and foot-and-mouth disease in a world of countries with diverse health statuses

Since 2001, with the exception of 2007, the UK has remained free of FMD, despite more than 90 countries in the world being considered endemic (officially 'non-free') [16]. Globally, there are seven regional FMD virus pools within which multiple, distinct virus genotypes evolve and circulate continuously within animal populations, providing a complex, ever-changing epidemic risk situation for FMD-free countries.

A global strategy to control FMD was launched in 2012, to reduce the daily burden of disease within FMD endemic communities and countries, and at the same time, reduce risks of regional pandemics spilling over to other continents and to free countries [58]. For endemic countries with limited resources and prospects for early eradication, impact reduction is recommended by targeting controls, such as vaccination, towards sectors that are most economically vulnerable to FMDV (e.g. improved but highly susceptible breeds of dairy cattle [59]). For FMD-free countries, an enormous effort with multiple safety controls is in place to retain their 'hard-won' health status. The key measures relate to controls on movements of animals and animal products, encompassing prohibitions and specific sanitary measures (e.g. surveillance, testing, quarantines, virus inactivation procedures), in compliance with international standards for safe trade [15].

In order to be considered FMD-free for international trade, countries have to submit evidence to the World Organisation for Animal Health (OIE) that they have the necessary control and surveillance measures in place to assure importers that their animals and products can be safely introduced. A range of options exists to facilitate international trade, such as attaining freedom with or without ongoing vaccination, according to the incursion risk from neighbouring countries. Countries can also establish disease-free zones (geographic areas) and compartments (managed areas separated by biosecurity practices) [15]. Since 2001, there has been a growing desire to establish more targeted measures (e.g. geographically, by use of sub-national containment zones, to deal with FMD incursions into formerly FMD-free regions), in order to mitigate the risk while minimizing the collateral damage caused by blanket restrictions, especially in relation to lost trade [16].

Biosecurity to protect the most vulnerable at-risk category currently appears to be the common ground between FMD and COVID-19 control in countries unable to eradicate the infection, to which herd immunity and targeted, risk-based vaccination may be added (if indeed protective) as the

recovered-immune fraction rises and a vaccine becomes available. With FMD, significant population immunity to at least one FMD virus serotype exists in most non-free regions [60] and is probably a key factor governing periodicity of epidemics at the regional scale [61]. In human health, herd immunity is a concept mostly associated with vaccination programmes, but in endemic situations for FMD an understanding of naturally acquired immunity can be used to target vaccination to sectors most at risk. These at-risk sectors also often maintain very strict biosecurity controls to reduce and prevent pathogen exposure. Similar approaches are likely to be maintained to keep COVID-19 out of at-risk groups, such as care homes.

For FMD, vaccination has limitations for protecting FMD-free countries from incursions, because of the need to keep revaccinating susceptible animals, and the serotype and sometimes strain-specific nature of vaccinal protection. Furthermore, animals and products from countries FMD-free with vaccination are perceived as higher risk compared to those from countries that are FMD-free without vaccination. If COVID-19 requires frequent revaccination or provides incomplete protection, then additional control measures are likely to be inevitable. With COVID-19, a divergence of strategies is already emerging between countries aiming at early eradication versus containment. The effectiveness of future vaccination will likely have a big impact on the long-term nature of these choices. As for FMD, there is a possibility of a world divided into COVID-19-free and endemic areas with status-related restrictions for international movements of people and certain commodities. There has already been talk of a COVID-19-free bubble encompassing New Zealand, Australia and perhaps other southern Pacific countries [62]. Future pre-travel testing and post-travel quarantine or monitoring might mirror that widely used for the control of veterinary diseases, where animals may have to be tested virologically and/or serologically to show the absence of infection. In some cases, pre-export testing for antibodies is used to substantiate safety for trade/movement by demonstrating immunity acquired by infection (e.g. bluetongue) or vaccination (e.g. rabies) [63]. With FMD, the economic impact of being seen as an infected or high-risk country is severe, and this can create a conflict between improving surveillance and transparency, and maintaining trade. Such conflicts would also appear with COVID-19 if reporting of cases is seen as reducing national prestige, tourism or personal freedoms to travel.

## 6. Conclusion

The speed and impact of the COVID-19 pandemic has led to a national and international response that is unprecedented in recent times. Animal health critical resources, including diagnostic laboratory capacities and protective equipment, have been diverted to support the COVID-19 response in many countries and major disease outbreak preparedness plans should include common and adaptable critical reserves (stockpiles) and capacities. The severity of the situation has led to the use of control measures that restrict individual freedoms whose use on this scale was not anticipated in many modern democracies, and where such approaches are typically reserved for the control of livestock diseases (electronic supplementary material, table S2). For the long-term control of COVID-19 or subsequent pandemic threats, there may be many local, national and international veterinary control measures that are relevant, for example, concerning movement between areas of different disease status. These could be adapted and applied during the current COVID-19 epidemic or in the long-term global control strategy.

Many lessons learnt from the UK 2001 FMD outbreak (table 1) are highly relevant to COVID-19 control. Under the 'One Health' paradigm, information and lessons learnt from experiences must be shared between the public health and veterinary fields. Both the betacoronaviruses (which include SARS-CoV-2) and FMDV have wildlife reservoirs and involve inter-species spill-overs and exchange driven by similar processes, with pandemic spread through globalized movements. Environmental, animal and human health are interconnected, and securing human health requires an understanding of the health of all parts.

Data accessibility. This article has no additional data.
Authors' contributions. K.S. drafted 70% of the main text. T.J.D.K.J. provided additional text. M.M. contributed the sections on UK 2001 transmission events, comparison of epidemic timelines and containment. D.J.P. drafted the tables and figure 1, and contributed text to all sections.
Competing interests. We declare we have no competing interests.
Funding. K.J.S. is funded by the EuFMD Commission. M.M. and D.J.P. are part funded by the EuFMD Commission through the EC (DG-SANTE) Phase V agreement on the EuFMD work programme. T.J.D.K.-J. is part funded through the CGIAR Research Program on Agriculture for Nutrition and Health (A4NH) and the GIZ ILRI One Health Research, Education, Outreach and Awareness Centre (OHRECA).

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
