## [Reviewer comments · Proceedings of the Royal Society B: Biological Sciences]

Review History

RSPB-2020-0906.R0 (Original submission)

Review form: Reviewer 1 (Marion Rowland)

Recommendation

Major revision is needed (please make suggestions in comments)

Scientific importance: Is the manuscript an original and important contribution to its field?

Good

General interest: Is the paper of sufficient general interest?

Good

Quality of the paper: Is the overall quality of the paper suitable?

Acceptable

Is the length of the paper justified?

No

Should the paper be seen by a specialist statistical reviewer?

No

Do you have any concerns about statistical analyses in this paper? If so, please specify them explicitly in your report.

No

It is a condition of publication that authors make their supporting data, code and materials available - either as supplementary material or hosted in an external repository. Please rate, if applicable, the supporting data on the following criteria.

Is it accessible?

N/A

Is it clear?

N/A

Is it adequate?

N/A

Do you have any ethical concerns with this paper?

No

Comments to the Author

This is a very interesting manuscript which is very detailed with a huge depth and range of information presented. To have its greatest impact for a general reader this reviewer feels that it needs to be a much shorter and concise document. The following are some suggestions

1. Table 1 is very effective. A number of Tables with comparisons between FMD and COVID19 under a number of different heading would greatly help the reader to understand the range of problem areas in the practice of One Health. A list of current data sources would also be helpful.
2. While the authors are very clear about the outcome of the inquiry into FMD2001, it is lost in the detail of the manuscript. The lessons learned, which can all be applied to COVID 19 should form a central theme of the manuscript and be included in the abstract.
3. The similarities in the lack of test capacity must be highlighted. While COVID 19 is considered a new virus, it is related to other SARS virus so the lack of investment in the development of accurate test capacity was not learned from FMD but must be learned for the next animal or human pandemic.
4. While the authors report the percentage of the total heard that were culled, some standardisation of the death rate such as the proportion of the population of England who died from COVID 19 rather than the actual numbers should be provided for a given time (April 30th 2020). Comparison of crude numbers, as currently presented in the media has little meaning.
5. A short section on the priorities for Public Health or One Health Preparedness, based on lessons learned from both Epidemics would also be helpful.
6. Table 2 is cited in the conclusion but I feel this is Table 1.

Review form: Reviewer 2

Recommendation

Accept with minor revision (please list in comments)

Scientific importance: Is the manuscript an original and important contribution to its field?

Excellent

General interest: Is the paper of sufficient general interest?

Excellent

Quality of the paper: Is the overall quality of the paper suitable?

Excellent

Is the length of the paper justified?

Yes

Should the paper be seen by a specialist statistical reviewer?

No

Do you have any concerns about statistical analyses in this paper? If so, please specify them explicitly in your report.

No

It is a condition of publication that authors make their supporting data, code and materials available - either as supplementary material or hosted in an external repository. Please rate, if applicable, the supporting data on the following criteria.

Is it accessible?

N/A

Is it clear?

N/A

Is it adequate?

N/A

Do you have any ethical concerns with this paper?

No

Comments to the Author

I enjoyed reading the article, which I found interesting and informative. I definitely recommend publication. However, I have a few remarks that should be considered in a revised version as follows:

- 1) The title should probably include reference to the "the UK"
- 2) Please compare estimates of R_0 for both diseases. Can you say anything about the severity of the diseases?
- 3) While COVID-19 spread rapidly in the UK following an exponential growth pattern, the 2001 FMD epidemics spread following a slower growth pattern in the number of premises infected (see <https://www.ncbi.nlm.nih.gov/pmc/articles/PMC6132414/>)
- 4) Please display the epidemic curves for both epidemics including a timeline of the key interventions

Review form: Reviewer 3

Recommendation

Major revision is needed (please make suggestions in comments)

Scientific importance: Is the manuscript an original and important contribution to its field?

Acceptable

General interest: Is the paper of sufficient general interest?

Good

Quality of the paper: Is the overall quality of the paper suitable?

Good

Is the length of the paper justified?

Yes

Should the paper be seen by a specialist statistical reviewer?

No

Do you have any concerns about statistical analyses in this paper? If so, please specify them explicitly in your report.

No

It is a condition of publication that authors make their supporting data, code and materials available - either as supplementary material or hosted in an external repository. Please rate, if applicable, the supporting data on the following criteria.

Is it accessible?

N/A

Is it clear?

N/A

Is it adequate?

N/A

Do you have any ethical concerns with this paper?

No

Comments to the Author

The manuscript "Foot-and-mouth disease in 2001 and COVID19 in 2020 : Parallels and differences in epidemic management" attempts to compare the characteristics of the 2001 foot-and-mouth disease epidemic with the current pandemic. I admit that my initial gut reaction to the abstract was a bit off-putting (I was afraid of a paper about COVID for the sake of COVID), but I found the paper an interesting read. Indeed, there are some similarities between the two: the presence of an aerosol-based infection pathway, the importance of testing for detection, and restriction of local, regional, and international movements. However, there are gross differences that make the comparison less compelling: many of the disease characteristics are quite different (e.g. FMD affects multiple species, with different levels of susceptibility and transmission routes, the presence of carrier species and wild reservoirs), sources of cost in the disease (FMD rarely kills - the main cost is in the trade response it induces), and obvious differences in management (we cannot cull people). Many of these are left out of the paper.

Because the paper switches between a UK and global focus, it is unclear to me what the primary research question is. If the primary research question is "What can we learn by comparing the current pandemic to the 2001 FMD epidemic?" then the paper must be revised to encompass the broader management strategies and incentives of FMD and COVID and provide a more balanced comparison of the two diseases. I do not need to compare COVID19 to FMD to know that early detection, testing, and movement restrictions are needed prevent spread and limit infections. But that is not to say that there are interesting lessons to be learned from the two. While some of the important details need to be added to more fully describe the 2001 epidemic, there is a story about the initial seeding and effectiveness of management efforts in regions that were

overwhelmed or not by the number of FMD cases. In addition, there are some interesting questions regarding the global management of FMD, and the trade offs associated with eradicating FMD in developing countries vs. welfare, that will be applicable to future COVID19 management.

If the primary research question is “Can we use the 2001 FMD epidemic to evaluate if the United Kingdom improvement its biosecurity and management policies for facing new epidemics?” then the paper is in better shape. That being said, much of the paper will need to be reworked and the authors will need to decide on when to focus on the UK versus the global story.

Regardless of the editorial decision, I believe that there need to be a number of changes to improve the manuscript. First, the scale of the study - particularly that of the 2001 epidemic - needs to be clarified. The authors flip back and forth between management within the United Kingdom and global management of FMD in 2001, which is confusing and changes what we can say about disease management. The World Organization of Animal Health maintains a set of five disease-free categorization ranging from disease-free, no vaccinations (low risk) to not disease-free (high risk). Countries with and without FMD face very different decisions in how they manage disease risk within their own countries but also on the global market. Questions of welfare and tradeoffs within developed and developing countries are important for FMD management (see the 2012 Global FMD Control Strategy) and will be for COVID19.

In this same vein, the nature of disease-free designations, and how they impact the different management strategies is important. Understanding why people choose their management decisions and how they are incentivized or compensated (maybe too well for the 2001 FMD epidemic) matters for the current management of an epidemic but also for how they will view and react to future risks of disease.

The temporal scales of the epidemics are also important. Where do each of the “phases” in the manuscript fall on the epidemic curves for each? Could the authors include a figure of each epidemic labeling the phases? There are some inconsistencies in the timing between the phases, such as testing centers being overwhelmed (which is in phase 1 but could easily fall in phase 2 and 3, no?).

Finally, if we are going to compare the two diseases, then it should be done so in a complete fashion. Many of the reasons that FMD is such a pain to manage are mostly ignored. Specifically, FMD infects virtually every cloven hoofed animal in the world, can be spread by multiple pathways with different animals exhibiting different susceptibilities and pathways of viral shedding, the presence of carrier animals, and wild disease reservoirs. While I understand that this will cause the comparison to diverge, it is important to remain transparent and open with the science of the two diseases.

I am including a number of other comments and suggestions embedded in the attached .pdf file. Please let me know if they are not viewable.

Decision letter (RSPB-2020-0906.R0)

27-May-2020

Dear Dr Sumption:

Your manuscript has now been peer reviewed and the reviewers’ comments (not including confidential comments to the Editor) are included at the end of this email for your reference. As you will see, the reviewers think the article has the potential to be a useful contribution but they

have raised some concerns with your manuscript that would need addressing first. Perhaps the most important are to clarify the scope (just UK or global?) and be more up-front about the (big) differences between the diseases – see referee 3's comments in particular. However, because the referees have been quite specific in their comments, and these look to me to be achievable, I would like to invite you to revise your manuscript to address them.

We do not allow multiple rounds of revision so I urge you to make every effort to fully address all of the comments at this stage. If deemed necessary, your manuscript will be sent back to one or more of the original reviewers for assessment. If the original reviewers are not available we may invite new reviewers. Please note that we cannot guarantee eventual acceptance of your manuscript at this stage. If the lessons are only for the UK and the differences between the diseases make extrapolation from FMD to COVID19 more tentative, then the case for publication in Proceedings B seems less compelling.

Research ethics:

Use of animals and field studies:

Please submit a copy of your revised paper within three weeks. If we do not hear from you within this time your manuscript will be rejected. If you are unable to meet this deadline please let us know as soon as possible, as we may be able to grant a short extension.

Best wishes,
Innes Cuthill

Prof. Innes Cuthill
Reviews Editor, Proceedings B
mailto: proceedingsb@royalsociety.org

Reviewer(s)' Comments to Author:

Referee: 1

Comments to the Author(s)

This is a very interesting manuscript which is very detailed with a huge depth and range of information presented. To have its greatest impact for a general reader this reviewer feels that it needs to be a much shorter and concise document. The following are some suggestions

1. Table 1 is very effective. A number of Tables with comparisons between FMD and COVID19 under a number of different heading would greatly help the reader to understand the range of problem areas in the practice of One Health. A list of current data sources would also be helpful.
2. While the authors are very clear about the outcome of the inquiry into FMD2001, it is lost in the detail of the manuscript. The lessons learned, which can all be applied to COVID 19 should form a central theme of the manuscript and be included in the abstract.
3. The similarities in the lack of test capacity must be highlighted. While COVID 19 is considered a new virus, it is related to other SARS virus so the lack of investment in the development of accurate test capacity was not learned from FMD but must be learned for the next animal or human pandemic.
4. While the authors report the percentage of the total heard that were culled, some standardisation of the death rate such as the proportion of the population of England who died

from COVID 19 rather than the actual numbers should be provided for a given time (April 30th 2020). Comparison of crude numbers, as currently presented in the media has little meaning.

5. A short section on the priorities for Public Health or One Health Preparedness, based on lessons learned from both Epidemics would also be helpful.

6. Table 2 is cited in the conclusion but I feel this is Table 1.

Referee: 2

Comments to the Author(s)

I enjoyed reading the article, which I found interesting and informative. However, I have a few remarks that should be considered in a revised version as follows:

- 1) The title should probably include reference to the "the UK"
- 2) Please compare estimates of R0 for both diseases. Can you say anything about the severity of the diseases?
- 3) While COVID-19 spread rapidly in the UK following an exponential growth pattern, the 2001 FMD epidemics spread following a slower growth pattern in the number of premises infected (see <https://www.ncbi.nlm.nih.gov/pmc/articles/PMC6132414/>)
- 4) Please display the epidemic curves for both epidemics including a timeline of the key interventions

Referee: 3

Comments to the Author(s)

The manuscript "Foot-and-mouth disease in 2001 and COVID19 in 2020 : Parallels and differences in epidemic management" attempts to compare the characteristics of the 2001 foot-and-mouth disease epidemic with the current pandemic. I admit that my initial gut reaction to the abstract was a bit off-putting (I was afraid of a paper about COVID for the sake of COVID), but I found the paper an interesting read. Indeed, there are some similarities between the two: the presence of an aerosol-based infection pathway, the importance of testing for detection, and restriction of local, regional, and international movements. However, there are gross differences that make the comparison less compelling: many of the disease characteristics are quite different (e.g. FMD affects multiple species, with different levels of susceptibility and transmission routes, the presence of carrier species and wild reservoirs), sources of cost in the disease (FMD rarely kills - the main cost is in the trade response it induces), and obvious differences in management (we cannot cull people). Many of these are left out of the paper.

Because the paper switches between a UK and global focus, it is unclear to me what the primary research question is. If the primary research question is "What can we learn by comparing the current pandemic to the 2001 FMD epidemic?" then the paper must be revised to encompass the broader management strategies and incentives of FMD and COVID and provide a more balanced comparison of the two diseases. I do not need to compare COVID19 to FMD to know that early detection, testing, and movement restrictions are needed prevent spread and limit infections. But that is not to say that there are interesting lessons to be learned from the two. While some of the important details need to be added to more fully describe the 2001 epidemic, there is a story about the initial seeding and effectiveness of management efforts in regions that were overwhelmed or not by the number of FMD cases. In addition, there are some interesting questions regarding the global management of FMD, and the trade offs associated with eradicating FMD in developing countries vs. welfare, that will be applicable to future COVID19 management.

If the primary research question is "Can we use the 2001 FMD epidemic to evaluate if the United Kingdom improvement its biosecurity and management policies for facing new epidemics?" then the paper is in better shape. That being said, much of the paper will need to be reworked and the authors will need to decide on when to focus on the UK versus the global story.

Regardless of the editorial decision, I believe that there need to be a number of changes to improve the manuscript. First, the scale of the study - particularly that of the 2001 epidemic - needs to be clarified. The authors flip back and forth between management within the United Kingdom and global management of FMD in 2001, which is confusing and changes what we can say about disease management. The World Organization of Animal Health maintains a set of five disease-free categorization ranging from disease-free, no vaccinations (low risk) to not disease-free (high risk). Countries with and without FMD face very different decisions in how they manage disease risk within their own countries but also on the global market. Questions of welfare and tradeoffs within developed and developing countries are important for FMD management (see the 2012 Global FMD Control Strategy) and will be for COVID19.

In this same vein, the nature of disease-free designations, and how they impact the different management strategies is important. Understanding why people choose their management decisions and how they are incentivized or compensated (maybe too well for the 2001 FMD epidemic) matters for the current management of an epidemic but also for how they will view and react to future risks of disease.

The temporal scales of the epidemics are also important. Where do each of the "phases" in the manuscript fall on the epidemic curves for each? Could the authors include a figure of each epidemic labeling the phases? There are some inconsistencies in the timing between the phases, such as testing centers being overwhelmed (which is in phase 1 but could easily fall in phase 2 and 3, no?).

Finally, if we are going to compare the two diseases, then it should be done so in a complete fashion. Many of the reasons that FMD is such a pain to manage are mostly ignored. Specifically, FMD infects virtually every cloven hoofed animal in the world, can be spread by multiple pathways with different animals exhibiting different susceptibilities and pathways of viral shedding, the presence of carrier animals, and wild disease reservoirs. While I understand that this will cause the comparison to diverge, it is important to remain transparent and open with the science of the two diseases.

I am including a number of other comments and suggestions embedded in the attached .pdf file. Please let me know if they are not viewable.

Author's Response to Decision Letter for (RSPB-2020-0906.R0)

See Appendix A.

RSPB-2020-0906.R1 (Revision)

Review form: Reviewer 1 (Marion Rowland)

Recommendation

Accept with minor revision (please list in comments)

Scientific importance: Is the manuscript an original and important contribution to its field?

Good

General interest: Is the paper of sufficient general interest?

Excellent

Quality of the paper: Is the overall quality of the paper suitable?

Good

Is the length of the paper justified?

Yes

Should the paper be seen by a specialist statistical reviewer?

No

Do you have any concerns about statistical analyses in this paper? If so, please specify them explicitly in your report.

No

It is a condition of publication that authors make their supporting data, code and materials available - either as supplementary material or hosted in an external repository. Please rate, if applicable, the supporting data on the following criteria.

Is it accessible?

Yes

Is it clear?

Yes

Is it adequate?

No

Do you have any ethical concerns with this paper?

No

Comments to the Author

(a) A population adjusted rate for the impact of SARS COV-2 rather than raw data for the number of deaths/infections/positive cases would greatly add to usefulness of the data for future pandemics. This is particularly important as testing has been so limited to date. Similarly a population based approach to the size of the cul for all animals is required.

(b) While the Data provided in Figure 2 is from other sources it would be helpful if the onset of each outbreak lined up at the start of the outbreaks

Review form: Reviewer 3

Recommendation

Accept with minor revision (please list in comments)

Scientific importance: Is the manuscript an original and important contribution to its field?

Good

General interest: Is the paper of sufficient general interest?

Good

Quality of the paper: Is the overall quality of the paper suitable?

Excellent

Is the length of the paper justified?

Yes

Should the paper be seen by a specialist statistical reviewer?

No

Do you have any concerns about statistical analyses in this paper? If so, please specify them explicitly in your report.

No

It is a condition of publication that authors make their supporting data, code and materials available - either as supplementary material or hosted in an external repository. Please rate, if applicable, the supporting data on the following criteria.

Is it accessible?

Yes

Is it clear?

Yes

Is it adequate?

Yes

Do you have any ethical concerns with this paper?

No

Comments to the Author

Overall, I find the paper much improved. It does a much better job at presenting a clear, up-front and accurate comparison of FMD and Covid-19, and the strengths/limitations on comparing them. My remaining concerns are primarily editorial, with several clarifications and typos. I move for acceptance pending (very) minor revisions.

Lines 17-18. The wording "... have experience of national scale epidemic crises of both diseases" is clunky. Reword to, "...have experienced epidemic crises on the national scale."

Lines 18-19. Reword to "... FMD and COVID-19, with respect to their management including..."

Line 43. In this section, you do not include a comparison to SARS-CoV-2. Consider renaming the header.

Lines 45-49. Please include the citations in the text. (They are already in the References section from Table 1, so including them here will not increase the word length of the paper.)

Line 48. Typo. Correct to "... and by ingestion, pigs being more readily..."

Line 49. Typo. Add such that "... to ruminants, which occurs..."

Lines 50-51. "... significant farm-to-farm spread can continue after imposition of complete animal movement standstills." It is worth noting that this depends on the spatial layout of farms. The fragmented farm landscapes characteristic of Europe is very different from the single, large tracks spread out and/or isolated in the United States system.

Line 59. Correct to "... mid-April 2001, and at the height ..."

Lines 51-52. Because you talk about the large costs of the disease after, it is worth briefly mentioning them here, e.g. the culling of over 2 million heads of livestock (Sobrino and Domingo, 2001) and total costs of about £6.5 billion (Thompson et al., 2002) (doi's below).

Sobrino and Domingo, 2001: 10.1093/emboreports/kve122

Thompson et al., 2002: 10.20506/rst.21.3.1353

Line 54. Typo. Should be, "... COVID19-associated..."

Line 76. Typo. Correct to, "... a disease-free status..."

Line 97. "Although the mechanisms behind the emergence of COVID19 are not yet understood..." Consider updating this statement. See Peter Daszak's work with EcoHealth Alliance. There are also several review papers that include possible origins (e.g. Harapan et al., 2020), as well as a Nature article that reviews the issue (doi: 10.1038/d41586-020-01541-z).

Harapan et al., 2020: 10.1016/j.jiph.2020.03.019

Lines 99-102. "One driver of wildlife consumption... which occurred immediately before COVID19 emergence." Consider deleting this sentence. We are fairly certain that Covid-19 came from bats, with an intermediary host between bats and people. This sentence does not contribute much and is, if anything, misleading of the complex process inherent to Covid-19.

Line 119. Change to, "... resources. Such models have been built..."

Line 125. Since this is a quote, add the reference number to the end of the sentence such that the text reads, "... reappraise their strategies and operations" [26]"

Line 138. "Information on the movements of livestock and where they are kept are routinely collected in the EU, especially for movement between countries." Could you add a reference where the reader could access information related to this?

Lines 138-141. Citation for this, please.

Lines 178-180. "... the EU member states cooperate... centrally agreed controls on movements of animals and animal products across borders." Could you direct the reader to what I assume is a WTO plan?

Line 181. Is "delay" the right word? Could you clarify/be a bit more specific? I am fairly certain that you mean "delay" [spread], but my mind went immediately to "delay" [introduction], which was confusing.

Line 195. Typo. Include a space such that "... in 2001 was a rationale..."

Line 219. Run-on sentence. Correct to, "... like COVID19. This is a challenge."

Lines 234 and 237. What do you mean by the phrases "the same periods of 2001" and "the denominators"? I do not understand. Please rephrase.

Line 242. Run-on sentence. Correct to, "... automatic culls. However, by then the scale..."

Line 264. It is worth pointing out the approximate percentage of the population that must be immune in order to achieve herd immunity (it should be >80%). This will greatly strengthen the argument of the "high burden of disease".

Line 289. You can cite Shanafelt and Perrings (2017) for a map/table of the global distribution disease-free/not disease-free countries for FMD.

Line 339. Typo. Rephrase to "... national crisis to plan for common, critical reserves..."

Line 341. Tweak to "... democracies, and where such approaches..."

Line 342. Add such that the paper reads, "... there may be many parallel local, national, and international veterinary..."

Lines 345-346. To be consistent, use quotations around "One Health".

Decision letter (RSPB-2020-0906.R1)

05-Oct-2020

Dear Dr Sumption

I am pleased to inform you that your manuscript RSPB-2020-0906.R1 entitled "Parallels, differences and lessons: a comparison of the management of foot-and-mouth disease and COVID19 using UK 2001/2020 as points of reference" has been accepted for publication in Proceedings B.

The referees are happy with the revisions and have recommended publication, but also suggest some further minor revisions. Therefore, I invite you to respond to the referees' comments and revise your manuscript. Because the schedule for publication is very tight, it is a condition of publication that you submit the revised version of your manuscript within 7 days. If you do not think you will be able to meet this date please let us know.

- 1) A text file of the manuscript (doc, txt, rtf or tex), including the references, tables (including captions) and figure captions. Please remove any tracked changes from the text before submission. PDF files are not an accepted format for the "Main Document".
- 2) A separate electronic file of each figure (tiff, EPS or print-quality PDF preferred). The format should be produced directly from original creation package, or original software format. PowerPoint files are not accepted.

3) Electronic supplementary material: this should be contained in a separate file and where possible, all ESM should be combined into a single file. All supplementary materials accompanying an accepted article will be treated as in their final form. They will be published alongside the paper on the journal website and posted on the online figshare repository. Files on figshare will be made available approximately one week before the accompanying article so that the supplementary material can be attributed a unique DOI.

Best wishes,
Innes

Prof. Innes Cuthill
Reviews Editor, Proceedings B
<mailto:proceedingsb@royalsociety.org>

Reviewer(s)' Comments to Author:

Referee: 1

Comments to the Author(s)

(a) A population adjusted rate for the impact of SARS COV-2 rather than raw data for the number of deaths/infections/positive cases would greatly add to usefulness of the data for future pandemics. This is particularly important as testing has been so limited to date. Similarly a population based approach to the size of the cul for all animals is required.

(b) While the Data provided in Figure 2 is from other sources it would be helpful if the onset of each outbreak lined up at the start of the outbreaks

Referee: 3

Comments to the Author(s)

Overall, I find the paper much improved. It does a much better job at presenting a clear, up-front and accurate comparison of FMD and Covid-19, and the strengths/limitations on comparing them. My remaining concerns are primarily editorial, with several clarifications and typos. I move for acceptance pending (very) minor revisions.

Lines 17-18. The wording "... have experience of national scale epidemic crises of both diseases" is clunky. Reword to, "...have experienced epidemic crises on the national scale."

Lines 18-19. Reword to "... FMD and COVID-19, with respect to their management including..."

Line 43. In this section, you do not include a comparison to SARS-CoV-2. Consider renaming the header.

Lines 45-49. Please include the citations in the text. (They are already in the References section from Table 1, so including them here will not increase the word length of the paper.)

Line 48. Typo. Correct to "... and by ingestion, pigs being more readily..."

Line 49. Typo. Add such that "... to ruminants, which occurs..."

Lines 50-51. "... significant farm-to-farm spread can continue after imposition of complete animal movement standstills." It is worth noting that this depends on the spatial layout of farms. The fragmented farm landscapes characteristic of Europe is very different from the single, large tracks spread out and/or isolated in the United States system.

Line 59. Correct to "... mid-April 2001, and at the height ..."

Lines 51-52. Because you talk about the large costs of the disease after, it is worth briefly mentioning them here, e.g. the culling of over 2 million heads of livestock (Sobrino and Domingo, 2001) and total costs of about £6.5 billion (Thompson et al., 2002) (doi's below).

Sobrino and Domingo, 2001: [10.1093/emboreports/kve122](https://doi.org/10.1093/emboreports/kve122)
Thompson et al., 2002: [10.20506/rst.21.3.1353](https://doi.org/10.20506/rst.21.3.1353)

Line 54. Typo. Should be, "... COVID19-associated..."

Line 76. Typo. Correct to, "... a disease-free status..."

Line 97. “Although the mechanisms behind the emergence of COVID19 are not yet understood...” Consider updating this statement. See Peter Daszak’s work with EcoHealth Alliance. There are also several review papers that include possible origins (e.g. Harapan et al., 2020), as well as a Nature article that reviews the issue (doi: 10.1038/d41586-020-01541-z).

Harapan et al., 2020: 10.1016/j.jiph.2020.03.019

Lines 99-102. “One driver of wildlife consumption... which occurred immediately before COVID19 emergence.” Consider deleting this sentence. We are fairly certain that Covid-19 came from bats, with an intermediary host between bats and people. This sentence does not contribute much and is, if anything, misleading of the complex process inherent to Covid-19.

Line 119. Change to, “... resources. Such models have been built...”

Line 125. Since this is a quote, add the reference number to the end of the sentence such that the text reads, “... reappraise their strategies and operations” [26]”.

Line 138. “Information on the movements of livestock and where they are kept are routinely collected in the EU, especially for movement between countries.” Could you add a reference where the reader could access information related to this?

Lines 138-141. Citation for this, please.

Lines 178-180. “... the EU member states cooperate... centrally agreed controls on movements of animals and animal products across borders.” Could you direct the reader to what I assume is a WTO plan?

Line 181. Is “delay” the right word? Could you clarify/be a bit more specific? I am fairly certain that you mean “delay” [spread], but my mind went immediately to “delay” [introduction], which was confusing.

Line 195. Typo. Include a space such that “... in 2001 was a rationale...”

Line 219. Run-on sentence. Correct to, “... like COVID19. This is a challenge.”

Lines 234 and 237. What do you mean by the phrases “the same periods of 2001” and “the denominators”? I do not understand. Please rephrase.

Line 242. Run-on sentence. Correct to, “... automatic culls. However, by then the scale...”

Line 264. It is worth pointing out the approximate percentage of the population that must be immune in order to achieve herd immunity (it should be >80%). This will greatly strengthen the argument of the “high burden of disease”.

Line 289. You can cite Shanafelt and Perrings (2017) for a map/table of the global distribution disease-free/not disease-free countries for FMD.

Line 339. Typo. Rephrase to “... national crisis to plan for common, critical reserves...”

Line 341. Tweak to “... democracies, and where such approaches...”

Line 342. Add such that the paper reads, “... there may be many parallel local, national, and international veterinary...”

Lines 345-346. To be consistent, use quotations around "One Health".

Author's Response to Decision Letter for (RSPB-2020-0906.R1)

See Appendix B.

Decision letter (RSPB-2020-0906.R2)

12-Oct-2020

Dear Dr Sumption

I am pleased to inform you that your manuscript entitled "Parallels, differences and lessons: a comparison of the management of foot-and-mouth disease and COVID19 using UK 2001/2020 as points of reference" has been accepted for publication in Proceedings B.

If you are likely to be away from e-mail contact during this period, let us know. Due to rapid publication and an extremely tight schedule, if comments are not received, we may publish the paper as it stands.

Open access

You are invited to opt for open access via our author pays publishing model. Payment of open access fees will enable your article to be made freely available via the Royal Society website as soon as it is ready for publication. For more information about open access publishing please visit our website at http://royalsocietypublishing.org/site/authors/open_access.xhtml.

The open access fee is £1,700 per article (plus VAT for authors within the EU). If you wish to opt for open access then please let us know as soon as possible.

Paper charges

Sincerely,
Editor, Proceedings B
<mailto:proceedingsb@royalsociety.org>

Appendix B

Response to referees

We appreciate the opportunity provided by the referees to improve our manuscript. The text has been redrafted, new tables and a new figure have been added. Clean and track-changed versions are supplied.

Listed below are the suggestions from each of the referees, with our responses in red text. The line numbering is cross-referenced to the new clean version of the manuscript.

Editor's comments

Clarify the scope (just UK or global?) and be more up-front about the (big) differences between the diseases.

We have now made clear in the title, abstract and the introduction that our intention is to look at similarities in the global management of these two diseases but using the UK responses to FMD in FMD 2001 and to COVID19 in 2020 as representative points of reference. We think that moving from the specific to the general is a good way to explore this topic.

One table (current Table S1) already showed similarities and differences in transmission characteristics between the viruses. We have added a section that compares the morbidity and mortality of FMD and COVID19 (supplementary text lines 31-48). We have added a new table to show similarities and differences with respect to control of FMD and COVID19 (Table S2).

Paper exceeds allowed length

As well as improving the conciseness of language, we have moved additional materials, including two tables to the supplementary materials in order to reduce the paper to 10 printed pages.

Referee #1

To have its greatest impact for a general reader this reviewer feels that it needs to be a much shorter and concise document.

Many of the referees' comments require the supply of additional information. To keep the paper as concise as possible, we have added some new materials including tables and moved these along with some of the original text to the supplementary materials.

The following are some suggestions

1. Table 1 is very effective. A number of Tables with comparisons between FMD and COVID19 under a number of different heading would greatly help the reader to understand the range of problem areas in the practice of One Health. A list of current data sources would also be helpful.

Table 1 (now Table S1) has had some extra information added on R_0 and the references have been strengthened. Two new tables and additional references have been added.

2. While the authors are very clear about the outcome of the inquiry into FMD2001, it is lost in the detail of the manuscript. The lessons learned, which can all be alineslied to COVID 19 should form a central theme of the manuscript and be included in the abstract.

A new Table (now Table 1) sets this out and provides references to the original reports. The key overall "one health" conclusion is now included in the abstract.

3. The similarities in the lack of test capacity must be highlighted. While COVID 19 is considered a new virus, it is related to other SARS virus so the lack of investment in the development of accurate test capacity was not learned from FMD but must be learned for the next animal or human pandemic

Additional details and emphasis added - lines 158-174; 254-268; 327-330; Tables 1 and S2.

4. While the authors report the percentage of the total herd that was culled, some standardisation of the death rate such as the proportion of the population of England who died from COVID 19 rather than the actual numbers should be provided for a given time (April 30th 2020). Comparison of crude numbers, as currently presented in the media has little meaning.

Calculations reworked - lines 63-64.

5. A short section on the priorities for Public Health or One Health Preparedness, based on lessons learned from both Epidemics would also be helpful

Included in abstract and conclusion - lines 22-24; 345-350

6. Table 2 is cited in the conclusion but I feel this is Table 1.

New Table numbering

Referee #2

1) The title should probably include reference to the "the UK"

See new title in which this is addressed.

2) Please compare estimates of R_0 for both diseases. Can you say anything about the severity of the diseases?

Both addressed - Table S1 and supplementary text lines 31-48

3) While COVID-19 spread rapidly in the UK following an exponential growth pattern, the 2001 FMD epidemics spread following a slower growth pattern in the number of premises infected (see <https://www.ncbi.nlm.nih.gov/pmc/articles/PMC6132414/>)

Now discussed - lines 188-190.

4) Please display the epidemic curves for both epidemics including a timeline of the key interventions

Done - see new Figure 2

Referee #3

However, there are gross differences that make the comparison less compelling: many of the disease characteristics are quite different (e.g. FMD affects multiple species, with different levels of susceptibility and transmission routes, the presence of carrier species and wild reservoirs), sources of cost in the disease (FMD rarely kills - the main cost is in the trade response it induces), and obvious differences in management (we cannot cull people). Many of these are left out of the paper.

These are now covered in greater detail (Tables S1 & S2, supplementary text, main text lines 32; 76; 296-308; Supplementary text lines 32-48), although it is not possible to keep the MS short and provide a textbook account of FMD. All of the specific points raised above (reservoirs, relative mortality, management differences such as culling) are now covered in text and/or supplementary tables and text.

Because the paper switches between a UK and global focus, it is unclear to me what the primary research question is. If the primary research question is "What can we learn by comparing the current pandemic to the 2001 FMD epidemic?" then the paper must be revised to encompass the broader management strategies and incentives of FMD and COVID and provide a more balanced comparison of the two diseases. I do not need to compare COVID19 to FMD to know that early detection, testing, and movement restrictions are needed prevent spread and limit infections. But that is not to say that there are interesting lessons to be learned from the two. While some of the important details need to be added to more fully describe the 2001 epidemic, there is a story about the initial seeding and effectiveness of management efforts in regions that were overwhelmed or not by the number of FMD cases. In addition, there are some interesting questions regarding the global management of FMD, and the trade offs associated with eradicating FMD in developing countries vs. welfare, that will be applicable to future COVID19 management

We think it is possible to cover two main themes: 1) were the relevant lessons of FMD in 2001 applied to COVID19 in 2020, and 2) is international control of FMD relevant to future COVID19 management.

If the primary research question is "Can we use the 2001 FMD epidemic to evaluate if the United Kingdom improved its biosecurity and management policies for facing new epidemics?" then the paper is in better shape. That being said, much of the paper will need to be reworked and the authors will need to decide on when to focus on the UK versus the global story.

We believe that we have addressed this research question and have reworked the paper in multiple ways to make the distinction between UK and global controls clearer. For example, by making clearer some of the different approaches needed for FMD control in endemic

versus FMD-free countries and regions. However, inevitable overlap remains between UK and wider responses to FMD and COVID19.

Regardless of the editorial decision, I believe that there need to be a number of changes to improve the manuscript. First, the scale of the study - particularly that of the 2001 epidemic - needs to be clarified. The authors flip back and forth between management within the United Kingdom and global management of FMD in 2001, which is confusing and changes what we can say about disease management. The World Organization of Animal Health maintains a set of five disease-free categorization ranging from disease-free, no vaccinations (low risk) to not disease-free (high risk). Countries with and without FMD face very different decisions in how they manage disease risk within their own countries but also on the global market. Questions of welfare and tradeoffs within developed and developing countries are important for FMD management (see the 2012 Global FMD Control Strategy) and will be for COVID19. In this same vein, the nature of disease-free designations, and how they impact the different management strategies is important. Understanding why people choose their management decisions and how they are incentivized or compensated (maybe too well for the 2001 FMD epidemic) matters for the current management of an epidemic but also for how they will view and react to future risks of disease.

We have reorganised the materials and headings and this makes it clearer where we are talking about the UK 2001 epidemic and wider control measures. Some further details are given of the international trade standards set for FMD by The World Organisation for Animal Health. The relevance of these measures to alternative future management options for COVID19 appears to have become more and not less compelling since we submitted our first draft.

The temporal scales of the epidemics are also important. Where do each of the “phases” in the manuscript fall on the epidemic curves for each? Could the authors include a figure of each epidemic labeling the phases? There are some inconsistencies in the timing between the phases, such as testing centers being overwhelmed (which is in phase 1 but could easily fall in phase 2 and 3, no?).

We have introduced a figure showing the epidemic curves and main events for each situation.

Finally, if we are going to compare the two diseases, then it should be done so in a complete fashion. Many of the reasons that FMD is such a pain to manage are mostly ignored. Specifically, FMD infects virtually every cloven hoofed animal in the world, can be spread by multiple pathways with different animals exhibiting different susceptibilities and pathways of viral shedding, the presence of carrier animals, and wild disease reservoirs. While I understand that this will cause the comparison to diverge, it is important to remain transparent and open with the science of the two diseases.

Not all knowledge about the biology and control of FMD can be summarised in this article. Nevertheless, we have included additional details about its host range, morbidity and mortality (in the supplementary materials) and about international FMD controls.

I am including a number of other comments and suggestions embedded in the attached .pdf file. Please let me know if they are not viewable.

Queries marked on text by referee #3 [NB *multiple queries added by referees as comments would have been easier to respond to in a word document than in a pdf file*]:

We have listed and replied to these (all but the most minor) in their original order and under the subheadings of the original text - see below. Due to reorganisation of the text and headings, in the new manuscript, the order may now have changed.

Abstract

What is the rate of emergence of new strains of FMD? **A simple answer cannot be given to this question and discussion would not be appropriate in the abstract.**

Change to disease-free regions. **Changed (line 12).**

I am not sure if you refer to the global FMD epidemic? **"UK" added to FMD 2001 (line 13)**

Introduction

In relation to economic impact of FMD - What, how much and why? **Added: due to production losses, costs of control and trade restrictions (line 32)**

Remove strong wording. **"disastrous" changed to "extensive" (line 35).**

Comparative viral and transmission characteristics

Aren't all viruses small? **Word "small" removed (line 45). Although, Picornaviruses are smaller than most (that's what Pico means).**

Queries about Table 1. **Revised Table 1 (now Table S1) and cited many original references.**

Queries about pathogenesis? **Revised statements about clinical signs and morbidity along with further details now provided in the supplementary text.**

Pathognomonic meaning. **word removed**

FMD2001 and COVID19: scale and impacts

Citation request. **Given: NAO (2002) (line 60)**

Difficulty in comparing proportions of infected farms and people. **Changed to statement of official UK SARS-COV-2 cases and deaths (line 63)**

Preparedness

Prevention of International Spread

Question in relation to Figure 1 about definition of FMD-free versus endemic areas? **Added footnotes to Figure 1**

Can authors briefly explain benefits of free status? **Sentence added: In the case of FMD, a free status is very important for international trade in susceptible livestock and their products, such as meat and milk (Shanafelt and Perrings, 2017) (line 308)**

Requested citation for data on incursions? **Provided as suggested (line 79)**

Query about whether all COVID incursions multifocal? **Qualified statement and added citation (line 80).**

Suggestions for changes to text about controls in place for movement of people..
Suggestions accepted and used (line 84)

Understanding and mitigating the key transmission events behind overwhelming outbreaks

Rephrasing of first sentence. **Done (line 91)**

Are post-movement standstills still in effect in UK? **Yes and sentence modified (lines 93-94)**

Change "but" to "although". **Done (line 95)**

Add citation. **Done (line 99)**

Query about risks and drivers associated with wildlife markets. **Qualified our statement that we are only describing one of the drivers. (line 99)**

Requested sentence to be rephrased. **Done**

Requested citation for flu pandemic contingency plan. **Not needed in support of existing statement (lines 125-6)**

Pre-epidemic phase - the warning signs

Query on final sentence. **Sentence removed.**

Resource planning

Query as to whether discussion on resource planning contributes to early warning?
Discussion moved to section on Preparedness planning. (lines 116-120)

Delayed reporting and detection: Missing the first cases

Invitation to say more about asymptomatic early spread of COVID19. **Not necessary to go into further details.**

Management of FMD 2001 and COVID19

Request for figure of the epidemic curves - **provided as new figure**

Comment that in UK in 2001 some local outbreaks were contained. **Agreed, but do not see the need to make this point.**

Request for more details on measures in UK and internationally. **New table provides more details on control measures.**

Query over statement about herd immunity as applied to FMD. **Moved word "initially" to make clear that this only refers to COVID19 and not FMD. (line 180) Agree with comment about challenge to achieve this without many deaths, in case of COVID19.**

Query over 4 day delay period - explained in full (lines 185-188).

Citation for current FMD strategy that refers to national standstill, Supplied (lines 190-1)

Objected to "desperate" ring culls. Changed to "aggressive" ring culls. (line 196)

Suggested change use of word "seeded". Sentence now removed.

Communication and the dangers of anticipation of imminent restrictions

Requested examples in support of statement about pre-announcement movements?
Sentence now removed

Query over "pre-lockdown" movements in Italy. Sentence now removed

Resources required to support the measures were overwhelmed

Citation request on portable tests developed in US. Provided (line 161)

Query over statement that: "The problems in the application of high throughout tests for FMD and point-of-care (POC) devices mirrors the issues with COVID19" Comment that "only recently have we had on-the-ground tests that do not need PCR in the lab. This is not correct as these tests were already developed in 2001.

Phase 2: the epidemic curves steepen

Consider renaming section - done (line 198)

Query over impact of surveillance effort - clarified by addition of word "reported" (line 199)

Citation suggestion - accepted (line 217)

Request to spell out traditional FMD control measures - done (line 203)

Clarify meaning of modelling groups - changed to "disease spread" modelling groups (line 206)

Expand on targets for the speed of slaughter and disposal - changed to on targets for the speed of culling operations (slaughter and disposal) (line 207)

Suggestion that more culled for disease versus welfare grounds. Yes, figures and citation added (lines 210-211)

Request for more examples to support statements on use of modelling ,including from influenza and ebola. Statements removed

Citation request in support of statement on local experience - provided along with parallels to COVID19 (lines 227-230)

Phase 3 the height of the battle

Reactions, community involvement and fightbacks

What is meant by battle of data - we think that this is made clear in subsequent sentences

Suggestion that confusion caused by switching between 2001 and 2020 epidemics. **This is necessary to make comparisons. Each statement now clearly references which epidemic is being discussed**

Suggestion regarding lack of testing and reporting - **added (line 238)**

When did the automatic culling stop - **sentence rephrased to link to 21st June 2001 (line 242).**

Phase 4: Exiting from the FMD2001 epidemic: comparisons

Flattening the curve

Long, medium and short term are not well defined. **Sentences removed**

Statement on preparedness of New Zealand in relation to COVID19 and FMD not clear. **Rewritten (lines 250)**

Citation requested for controls on bovine TB and bluetongue. **This section now modified for greater precision and referenced (lines 329-331)**

Query relevance of rabies testing. **Rabies is a good example of a veterinary zoonotic measure to safeguard international movement. See above.**

Various queries in relation to use of modelling for exit strategies from COVID 19. **This section now removed.**

Request to explain that recognition of freedom required for trade - **added this to sentence. Plus further explanation of OIE role provided later in section on international movements (lines 292-299).**

Request for citations on POC testing and difficulties with establishing automated serology. **POC testing citations provided elsewhere and references to problems with automated serology removed.**

Phase V: Reconstruction and long term impacts

Query about significance of impacts of FMD on human health. **We do not say that the impacts were equivalent, only that they were substantial and provide examples and citations.**

Not to mention hundreds of thousands of people dying. **Added "As well as the many deaths" (line 272-3)**

Request to expand on non-COVID health impacts. **These are obviously worthy of attention, but we do not think this paper is the right place for this.**

Phase 6: aftermath of FMD2001 and COVID19: coping with a world of countries with diverse health status

Suggest to change subtitle and change word "coping". **New subtitle: "The long term management of COVID19 and FMD in a world of countries with diverse health status" (lines 286-7)**

Word "fresh" removed. Done (line 291)

Who launched the global FMD strategy? Citation given (line 294)

Request for more details on benefits for FMD endemic country and controls for free countries- additional details provided (lines 294-299)

Request for examples of how countries without status can trade with free countries. Added new text: (lines 300-308)

Question about how movement of people can be managed geographically in ongoing global pandemic. This section now removed

Query over herd immunity discussions on COVID and FMD. Qualifying caveat added concerning COVID immunity (may not be protective) (line 311). Redrafted statements on FMD immunity which is already supported by citations (lines 311-317).

Challenge on explanation of non-vaccination use in free countries. Subtleties now explained in more detail (lines 318-321).

Conclusions

Challenge on how unusual has been use of quarantines in human disease control. Added caveat about scale of use for COVID19 (line 340)

Question regarding validity of any comparison between transmission characteristics of cows and people. Seems an extreme position, although context is all important and we have tried to provide this.

Consideration of term veterinary versus livestock. It is not unreasonable to mention wider veterinary diseases and their control.

Questions discussion over cross-species transmission. It is true that following the initial incursion, the COVID19 epidemic has been all about person-to-person spread. When FMD goes from cape buffalo to cattle, the same thing happens. But this does not mean that the wildlife reservoirs are not important.

Question over how compelling is the case for ASF influence on desire for other forms of meat. Suggests should be removed. Removed from conclusions